# Impact of Long-Term Cyclamate and Saccharin Consumption on Biochemical Parameters in Healthy Individuals and Type 2 Diabetes Mellitus Patients

**DOI:** 10.3390/medicina59040698

**Published:** 2023-04-03

**Authors:** Husni Mohammed Hasan, Suad Yousif Alkass, Daniele Suzete Persike de Oliveira

**Affiliations:** Department of Medicinal Chemistry, College of Pharmacy, University of Duhok, Duhok 1006, AJ, Iraq

**Keywords:** type 2 diabetes mellitus, artificial sweetener, saccharin, cyclamate, oxidative stress, atherosclerotic effect, metabolic dysfunction

## Abstract

Background: Previous studies on saccharin and cyclamate were either limited to experimental animals or lacked evaluation of their long-term consumption effects in humans. Objectives: This study evaluated the effect of chronic consumption of saccharin and cyclamate on biochemical parameters in healthy individuals and patients with type 2 diabetes mellitus. Material and Methods: Healthy and diabetic individuals were classified into two groups based on whether they consumed sweeteners or not. The participants were classified according to the amount of sweetener consumed per day and duration of consumption. Serum catalase activity, peroxynitrite, ceruloplasmin, and malondialdehyde concentrations were determined. Glycated hemoglobin, fasting glucose, creatinine, alanine transaminase, and lipid profile were also evaluated. The results suggest that saccharin and cyclamate increased HbA1C (+11.16%), MDA (+52.38%), TG (+16.74%), LDL (+13.39%), and TC/HDL (+13.11%) in healthy volunteers. Diabetic patients consuming sweeteners showed increased FSG (+17.51%), ceruloplasmin (+13.17%), and MDA (+8.92%). Diabetic patients showed a positive correlation between the number of tablets consumed per day with FSG and serum creatinine. A positive correlation was found between the duration of sweetener consumption and FSG as well as TG. Conclusion: Consumption of saccharin and cyclamate affected biochemical parameters related to metabolic functions in a time and dose-dependent manner and appear to increase oxidative stress in healthy and diabetic type 2 patients.

## 1. Introduction

In the context of current theories of the developmental origins of adult diseases, the patterns of change in dietary intake and energy expenditure related to the global nutrition transition are of great importance [1]. With increased consumer interest in reducing sugar intake, food products made with sweeteners rather than sugar have become more common [2]. Obese and diabetic patients, in addition to individuals concerned about their diet, are more likely to use sugar-free and low-calorie products to reduce calorie consumption and control blood glucose levels [3]. Several low-calorie synthetic sweeteners have recently emerged in the pharmaceutical and food industries, but their health risks due to their side effects restrict their adoption [4,5]. Several previous studies have shown that the use of artificial sweeteners can be hazardous to human health [6,7].

Aspartame, saccharin, acesulfame-K, and cyclamate have become sugar alternatives to replace sucrose [8], and have been widely used in dairy products, energy control diets, and diabetes management in Africa, Asia, Europe, and the US [7].

The main objective of diabetes management is controlling blood glucose; therefore, the food industry has invested heavily in research to discover potential alternative sweeteners [9]. More than half of the population prefers artificially sweetened food [10].

According to Zeynep and Sifa (2014), sweeteners may cause a variety of health problems, including cancer. Their arguments are based on studies dating to the 1970s, which linked saccharin to bladder cancer in laboratory rats. Because of these studies, saccharin previously carried a warning label stating that the product might be hazardous to human health [11].

It was shown that the consumption of a solution sweetened with saccharin promoted a greater total caloric intake and was associated with increased body weight and adiposity in rats, in a similar manner to the consumption of a solution sweetened with glucose [12].

The close association between oxidative stress and lifestyle-related diseases has become well-known [13]. The oxidative stress produced by saccharin may explain the role of this sweetener in the development of carcinoma. This effect is mainly observed when the sweetener is accumulated in a high dose in the bladder, or other tissues, as shown in reports of hyperplasia induction [7]. Significant inhibition of the antioxidant system was shown in rats’ livers during saccharin administration compared to control animals. Catalase and SOD activities were inhibited, as well as glutathione activity, which prevented cell death by toxic radicals [14].

Significant elevation of serum creatinine and urea concentrations with both high and low doses of saccharin also have been reported previously [14,15].

The compounds most commonly used on the market today as substitutes for common sugar in diet drinks and food products are saccharin, aspartame, and cyclamate. Many fears have arisen around the associated side effects of consuming artificial sweeteners in addition to broader concerns about their safety for human consumption.

The studies previously conducted on saccharin and cyclamate were limited to experimental animals. No previous study has described the effects of chronic consumption of the cyclamate and saccharin combination on oxidative stress, lipid profile, glycemic control, creatinine, and alanine transaminase activity in healthy individuals and in patients with type 2 diabetes mellitus (T2DM).

Considering the lack of information regarding the chronic use of artificial sweeteners in humans, the main goal of the present study was to evaluate the effect of chronic consumption of saccharin and cyclamate on biochemical parameters in healthy individuals and patients with type 2 diabetes mellitus.

## 2. Materials and Methods

### 2.1. Study Design and Sampling

All experiments had the approval of the Ethics Committee of the Directorate of Health-Duhok and the Postgraduate Committee of the College of Science (16092015-11-1). All methods were performed in accordance with the guidelines and regulations.

The cross-sectional study was carried out at the Diabetic Center of the General Azadi Teaching Hospital of Duhok in the Kurdistan Regional Government of Iraq from July 2015 to October 2016.

A total of 216 diabetic outpatients were informed about the nature of the study which was followed by an invitation to participate in the study.

Healthy volunteers (107) were selected from relatives of the University of Duhok staff, as well as hospital employees and their outpatients.

Informed written consent was obtained from each subject involved in this study after explaining the nature of the study. A pre-tested questionnaire was designed for matching the study needs. It included obtaining information on age, gender, smoking habits, alcohol consumption, past medical history (heart, liver, kidney, and other diseases), duration of diabetes mellitus, and family history of diabetes mellitus in addition to type, amount, and duration of sweetener consumption.

The exclusion criteria were patients with T1DM, gestational diabetes, cancer, liver diseases, heart diseases, renal diseases, endocrine disorders, patients on supplements, or smokers and alcohol consumers.

### 2.2. Study Groups

After the exclusion of 60 patients according to the exclusion criteria mentioned above, the 181 remaining patients with T2DM and the 82 healthy individuals with an age range between 35 and 70 were enrolled in this study.

The participants were further subdivided according to the presence of consumption of saccharin and cyclamate mixture in tablets.

Of the 181 diabetic patients, 88 were patients who consumed sweeteners (DS) and the remaining 93 individuals were patients who did not consume sweeteners (D). The 82 healthy individuals were divided into two groups, where 14 individuals were those who consumed sweeteners (HS) and the 68 remaining individuals who did not consume sweeteners (H).

Diabetic patients were also classified according to the amount of sweetener consumed per day into three groups: <5, 5–10, and >10 tablets. The mixture of cyclamate and saccharin per tablet contains 40 mg of cyclamate-sodium (E 952) and 4 mg of saccharin-sodium (E 954) (10:1 mixture). This way, <5 tablets represent less than 200 mg of cyclamate and 20 mg of saccharin. Between 5 and 10 tablets represent from 200 to 400 mg of cyclamate and from 20 to 40 mg of saccharin. Above 10 tablets represent more than 400 mg of cyclamate and more than 40 mg of saccharin. All sweetener-consuming participants used doses within the ADI for cyclamate and saccharin.

Besides that, diabetic patients were classified according to the duration of sweetener consumption into three groups: <5, 5–10, and >10 years of sweetener consumption.

### 2.3. Collection of Samples

Venous blood samples (10 mL) were collected in Duhok Diabetic Center from the 181 T2DM patients (Diagnosed according to the WHO protocol) and from the 82 healthy individuals using a disposable syringe after 12 to 14 h of fasting. Blood pressure was measured, and body mass index (BMI) and waist circumference (WC) were calculated.

The whole blood (2 mL) was placed immediately into a lavender top tube containing EDTA for the estimation of HbA1c. The remaining (8 mL) was used to obtain serum to analyze all other parameters. Measurements for HbA1C, serum glucose, lipid profile, ALT, and creatinine were performed manually using a BIOLABO kit on the same day as the sample collection. The remaining serum was divided into 5 parts in Eppendorf tubes and frozen at −28 °C until the oxidative stress parameters were measured.

### 2.4. Methods

To evaluate the adverse effects of saccharin and cyclamate mixtures on biochemical parameters in blood, enzymatic methods were applied to measure the serum level of glucose [16], TC [17], and TG [18]. The level of HDL was measured according to Burtis (1999) [19]. LDL concentration was calculated according to the Friedewald formula [20] as follows:
LDL-C (mg/dL) = TC − (HDL-C + VLDL-C)

Glycated Hemoglobin (HbA1c) was estimated by Nephelometry (GENIUS) immediately after sample collection. Serum creatinine level was assayed according to Jaffe’s Method using alkaline picrate reagent [21] and ceruloplasmin (CP) level was estimated according to the modified Menden Method using P-phenylenediamine [22]. MDA levels were assessed spectrophotometrically by measuring the pink chromogen compound produced from coupling between MDA and thiobarbituric acid [23,24]. Peroxynitrite radical was assayed by nitration of phenol. The absorbance of nitrophenol product was measured at 412 nm [25].

The activity of ALT was assayed according to the Reitman and Franke method [26], whereas the activity of catalase was estimated according to Hadwan and Abeds [27].

All the reagents and chemicals used in these experiments were analytical grade. The measurements were performed at the College of Science at the University of Duhok.

The normality of the values was checked prior to the comparisons between the groups.

### 2.5. Statistical Analysis

Statistical Package for Social Science (SPSS) software version (23.0) was used to analyze all the data. Data are expressed as mean ± Standard Deviation (SD). All the study groups were analyzed by Skewness and Kurtosis Test regarding their normal distribution before proceeding with the comparisons between the groups. *p* value < 0.05 was considered statistically significant for all analyses. One-way ANOVA was used to compare mean values between the groups. An independent t-test was used to compare the percentage of changes that occurred between the sweetener consumers group versus non-consumers. Pearson’s correlation coefficients were used to evaluate the association between several biochemical parameters in diabetic patients and increased amount of sweetener consumption.

## 3. Results

### 3.1. Effect of Long-Term Consumption of Artificial Sweeteners on Anthropometric and Biochemical Parameters in Diabetic Patients

The study found a significant increase in fasting blood glucose (FSG), ceruloplasmin, and malondialdehyde (MDA) levels in addition to increased activity of serum catalase in diabetic patients who were sweetener consumers (DS) compared with diabetic patients who were not sweetener consumers (D) (Table 1).

For healthy individuals, the study showed that saccharin and cyclamate mixtures significantly increased HbA1C and MDA levels (12.08 and 56.10%) and decreased the ceruloplasmin level, as well as, the catalase activity in HS compared to H (−30.83 and −39.61%) (Table 2 and Table 3).

A comparison between different groups as shown in Figure 1 revealed that WC, BMI, systolic, and diastolic blood pressure presented a significant increase when diabetic individuals with (DS) or without sweetener consumption (D) were compared to the healthy group (H) (*p* < 0.001). Whereas a comparison of the diabetic with sweetener (DS) group to the diabetic group without sweetener consumption (D) presented non-significant changes in WC, BMI, systolic and diastolic blood pressure in addition to ALT and creatinine level.

Serum fasting glucose and HbA1C results presented a significant increase (112.81 and 64.79%) in group D when compared to group H (*p* < 0.001). Further significant increases in FSG (17.51%) were observed in the DS group as compared to the D group (Figure 2).

#### 3.1.1. Effect of Long-Term Artificial Sweetener Consumption on Lipid Profile

Our study found that the levels of TC, TG, LDL, and TC/HDL increased significantly, whereas the level of HDL decreased in diabetic patients long-term sweetener consumers (DS) relative to the group healthy (H). Whereas a comparison of DS with the D group has shown a non-significant variation in lipid profile panel tests (Figure 3).

#### 3.1.2. Effect of Long-Term Artificial Sweetener Consumption on Oxidative Stress Biomarkers

Regarding the oxidative stress parameters, the results showed that serum malondialdehyde (MDA), peroxynitrite, and ceruloplasmin levels were significantly increased by 29.88, 40.31 and 5.67%, respectively, while serum catalase activity was inhibited (−25.44%) in the diabetic (D) group as compared to the healthy group (H). Further significant increases in MDA and ceruloplasmin levels (9 and 13%) and a decline in catalase activity (−12%) were observed when the diabetic patients long-term sweetener consumers (DS) group was compared to the D group, reflecting the effect of long-term sweetener consumption (Figure 4).

#### 3.1.3. Effect of the Amount of Sweetener Consumed Daily by Diabetic Patients on Anthropometric and Biochemical Parameters

A comparison of data obtained from diabetic patients who consumed less than 5 tablets of sweetener per day versus the group who consumed between 5 and 10 tablets daily revealed significantly increased FSG and serum creatinine levels of 23 and 17%, respectively, in the group who consumed 5-10 tablets (Table 4).

The group who consumed less than 5 tablets when compared to those who consumed > 10 tablets daily revealed increased levels of FSG and serum creatinine of 15 and 14%, respectively, in the group who consumed >10 tablets. A comparison between the same groups mentioned above showed a significant decrease in serum ceruloplasmin (−13; less than 5 tablets compared to 5–10 tablets and −26%; less than 5 tablets compared to >10 tablets) and serum peroxynitrite levels (−3; less than 5 tablets compared to 5–10 tablets and −31%; less than 5 tablets compared to >10 tablets), besides a decrease of serum catalase activity (−8; less than 5 tablets compared to 5–10 tablets and −28%; less than 5 tablets compared to >10 tablets).

Using the Pearson correlation coefficient^®^, a positive and significant correlation was found between sweetener consumption and FSG (r = 0.217, *p* = 0.042) as well as serum creatinine (r = 0.267, *p* = 0.012). Whereas inverse correlations were found between sweetener consumption and serum catalase activity (r = −0.361, *p* = 0.001), serum peroxynitrite (r = −0.239, *p* = −0.025), and serum ceruloplasmin (r = −0.262, *p* = 0.014) as shown in (Figure 5).

#### 3.1.4. Effect of the Duration of Daily Sweetener Consumption

When the data of diabetic patients who consume sweeteners were analyzed based on how long they have consumed sweeteners (Table 5), it showed a significant increase in the levels of TG (48)%, LDL-C (16)%, and TC/HDL (24)% in diabetic patients who consumed sweeteners for more than 10 years when compared to patients consuming sweetener for less than 5 years (Table 5).

Using the Pearson correlation coefficient (r), a positive and non-significant correlation was found between BMI and HbA1c (%) (r = 0.132, *p* = 0.653) in healthy individuals sweetener consumers (Figure 6).

A positive correlation was found between the period of sweetener consumption and the levels of FSG (r = 0.22, *p* = 0.04) and serum TG (r = 0.262, *p* = 0.014) (Figure 7).

## 4. Discussion

### 4.1. Effect of Artificial Sweeteners Consumption on Anthropometric Measurements

Sweetness, if not conjugated with calorie production, results in ambiguous psychobiological signals that confuse the body’s regulatory mechanisms, leading to a loss of control over appetite and overeating [28,29]. As a result, intense sweeteners have been blamed for the obesity epidemic [29,30].

Animal studies have convincingly proven that artificial sweeteners cause body weight gain. A sweet taste induces an insulin response, which causes blood sugar to be stored in tissues, but because blood sugar does not increase with artificial sweeteners, there is hypoglycemia and increased food intake. So, in the experiment, after a while, rats given artificial sweetener steadily increased caloric intake, resulting in increased body weight and adiposity. [31,32,33]. The above-suggested mechanisms were considered compatible with the results obtained in the current study. In the present study, healthy and diabetic patients who reported using saccharin and cyclamate mixtures had higher average BMI and WC, and a higher prevalence of abdominal obesity than participants who never reported using low-calorie sweeteners. Similarly, no variation in BMI was observed when the amount and consumption frequency increased.

A National Health and Nutrition Examination Survey analysis of adolescents showed a positive and linear association between sugar-sweetened beverage consumption and blood pressure [34,35]. A potential explanation for the relationship between artificially sweetened beverages and hypertension risk is the beverages’ association with the development of metabolic disturbances that in turn might lead to elevated blood pressure [36].

Similar results were obtained in the present study, since systolic and diastolic blood pressure were slightly and non-significantly increased when healthy and diabetic patients who consume saccharin and cyclamate mixtures were compared to those who did not consume the same mixtures. The increase in systolic blood pressure (3%) was more than that observed in the diastolic blood pressure.

### 4.2. Effect of Artificial Sweeteners Consumption on Serum Glucose and Glycated Hemoglobin

It was previously shown that the acceptable daily intake (ADI) of artificial sweeteners such as saccharin and cyclamate has no significant effect on glycemic control or blood lipids in persons with diabetes [37].

Previous studies have found a positive correlation between artificial sweetener consumption and several metabolic-syndrome-related clinical parameters such as increased weight and waist-to-hip ratio, higher fasting blood glucose, elevated glycosylated hemoglobin (HbA1C%), and glucose intolerance. Serum alanine aminotransferase was also found to be elevated [38].

A significant positive correlation was found between sweetener consumption and metabolic syndrome-related clinical parameters such as elevated FSG and HbA1c levels in 381 non-diabetic individuals [39].

In healthy volunteers, blood glucose slightly increased, whereas HbA1C increased significantly (11.15%) when sweetener consumers were compared to non-consumers. Similar results were found in diabetic patients. The unexpected result was obtained by comparing diabetic sweetener consumers with diabetic non-consumers which led to a significant increase in blood glucose by 17.51%. Artificial sweetener consumption is correlated with insulin resistance, incidence of type 2 diabetes, and poor glucose control in patients with pre-existing diabetes [40]. It is well known that the sweet taste induces an insulin response, which causes blood sugar to be stored in tissues, but because blood sugar does not increase with artificial sweeteners, there is hypoglycemia and increased food intake [31,32,33]. This mechanism can help explain why glucose and HbA1c levels increase as a result of sweeteners consumption.

The effect of sweeteners consumption on glycemic status as mentioned above came in line when the effect of the number of sweeteners consumed daily, and the duration of sweetener consumption by diabetic patients were tested. Significant increases in blood glucose were observed from 196.96 to 242.5 and to 230.44 mg/dL as the daily consumption of sweeteners increased from <5 to 5–10 and to >10 tablets/day, respectively. A gradual but not significant increase in HbA1C also was observed from 7.98 to 8.53 and to 8.85% as the consumption of sweeteners increased from <5 to 5–10 and finally to >10 tablets/day, respectively.

Similarly, when the effects of consumption frequency were examined, a gradual but not significant increase was observed in blood glucose from 211.82 to 228.38 and to 263.6 mg/dl, as well as HbA1C from 8.07 to 8.65 and to 8.8% as the period of consumption extended from <5 to 5–10 and to >10 years, respectively.

To the best of our knowledge, this is the first study that could add clear evidence and support a growing body of literature on the effects of the consumption of artificial sweeteners on glycemic control. In addition, we also explored additional adverse effects of prolonged artificial sweetener consumption on parameters related to metabolic syndrome, and the performance of the liver and kidneys.

### 4.3. Effect of Artificial Sweeteners Consumption on the Liver and Renal Functions

Previous results have shown that saccharin harmfully altered biochemical markers of liver and kidney function at both higher and lower doses [14]. Our group, in agreement with other studies, has previously shown that rats who ingested either low or high doses of saccharin exhibited significantly increased activity of serums ALT, AST, and ALP, which is a common sign of impaired liver function [7,41,42]. Other studies have found no significant difference in ALT and ALP activity in saccharin-treated rats compared to the control group [43].

Chronic consumption of saccharin may cause kidney injury according to the results obtained by [42], who showed a significant elevation in creatinine levels, which could be a result of reduced glomerular filtration followed by retention of urea and creatinine in the blood [44].

The present study showed that ALT activity increased in HS (18%) and to a lesser extent in DS, but both differences were not significant as compared to the H group. Similar results were also obtained for serum creatinine, where a non-significant increased level (7%) was found in healthy consumer subjects. Furthermore, no variation occurred in serum creatinine levels in diabetic patients. No effect on ALT activity was observed as a result of increasing the amount of sweetener consumed, nor increasing the frequency of consumption. Whereas, in diabetic patients, the serum creatinine level was increased significantly from 0.76 ± 0.19 to 0.9 ± 0.19 and to 0.87 ± 0.25, as the amount of the sweeteners consumed daily was increased from <5 to 5–10 and to >5 tablets/day, respectively. The frequency of sweetener consumption did not appear to have significant effects on overall creatinine levels.

### 4.4. Effect of Artificial Sweeteners Consumption on Lipid Profile

Concerning lipid metabolism, previous results have demonstrated that triglycerides and total cholesterol levels decreased in response to oral administration of saccharin in rats [41]. Both low and high doses of saccharin can induce hypocholesterolemia. High saccharin doses appear more likely to induce hypotriglyceridemia when compared to control groups [45,46]. Osfor and Elias reported those same effects after 12 weeks of saccharin consumption [47].

Long-term consumption of artificial sweeteners might accelerate atherosclerosis and senescence via impairment of the function and structure of apoA-I and HDL as shown by Jang et al. [48]. It was suggested that long-term consumption of artificial sweeteners may modify apoA due to the production of advanced glycated end (AGE) products [48] and protein cleavage, which is associated with loss of antioxidant ability and impairment of phospholipid binding ability, even in low concentrations [49].

Modification of apoA-I including cleavage, oxidation, nitration, and chlorination, could lead to the production of dysfunctional apoA and HDL [50,51]. These results suggest that all artificial sweeteners, especially saccharin, could cause modification of tertiary structures making it easier for proteolytic attacks to take place in serum, [52]. Superoxide anion facilitates oxidative modification of low-density lipoprotein (LDL) which plays a key role in the formation of atherosclerotic lesions [53].

The results obtained in the current study add further evidence of atherosclerotic effects in both diabetic and non-diabetic patients induced by consuming artificial sweeteners such as saccharin and cyclamate.

Consumption of sweeteners in healthy individuals led to a significant raise in triglyceride (16.74%), LDL (13.39%), and TC/HDL (13.11%). In contrast, HDL levels fell significantly by 7.13%. Additional studies where the diet of the participants is under control, including their consumption of sweetened foods and beverages are still needed. Overall, sweetener consumption by diabetic patients had a negligible effect on lipid components.

Regarding the effect of the number of sweetener tablets consumed by diabetic patients, a decrease in the levels of total cholesterol, LDL, and TG was noticed when the number of sweeteners consumed increased from <5 to 5–10 and to >5 tablets/day. These results affirm results from previous studies involving experimental diabetic rats. The lipid components were significantly affected by a prolonged period of artificial sweetener consumption, while the level of TG significantly decreased. LDL and TC/HDL levels increased significantly when the duration of sweetener consumption increased from <5 to 5–10 and to >10 per year. We believe our data presents a novel dimension by being the first attempt to study the effect of chronic saccharin and cyclamate consumption in humans.

### 4.5. Effect of Artificial Sweetener Consumption on Biomarkers of Serum Oxidative Stress

In a previous study, saccharin was shown to have increased ROS production at basal and maximal glucose levels and decreased cell respiration in a dose-dependent manner [54]. A significant inhibition of the antioxidant defense systems has been shown during saccharin administration, specifically a decrease in catalase, SOD, and GSH activities, which prevents cell death caused by the production of toxic radicals [7,14]. In contrast, MDA levels increased as a product of lipid peroxidation as a result of ROS action on lipids of cellular membranes [7,14].

The oxidative stress induced by high doses of saccharin has been attributed to inflammation initiated by liver cells. Stimulated inflammatory cells undergo a respiratory burst and release ROS, such as superoxide anion and hydrogen peroxide [55]. Peroxynitrite is a product of the controlled reaction of superoxide with nitric oxide that is capable of oxidizing biomolecules in a fashion similar to the hydroxyl radical [56].

All of the above-mentioned studies were carried out on animals and an intensive review of previous literature showed previous data on humans was not available, consequently, the data obtained in the present study can be considered original.

Peroxynitrite decreased (−10%) (*p* < 0.05) in the healthy and diabetic group (−9%) who consumed sweeteners relative to non-consumers. Previous works by [57,58] have suggested that peroxynitrite was consumed as a potent oxidizing and nitrating species.

In diabetic patients, peroxynitrite levels decreased gradually from 1.74 to 1.69 and to 1.2 mmol/L as the number of sweeteners consumed from <5 to 5–10 and, finally, to >10 tablets/day, respectively. The prolonged period of sweetener consumption for more than 10 years exacerbated the oxidative status and increased peroxynitrite levels in excess of an amount sufficient to exert its effect as a nitrating and oxidizing species.

MDA levels were significantly elevated by 52.38% in healthy human volunteers and increased (9%) in diabetic patients when the results of sweetener consumers were compared to non-consumers. These results indicate that the increase in lipid peroxidation results from ROS activity on lipids in the cellular membrane. No effect was observed as a result of increasing the amount and the duration of sweetener consumption by diabetic patients on MDA.

Ceruloplasmin, a copper-carrying metalloenzyme, acts as an antioxidant through its ferroxidase activity [59]. However, in conditions of elevated oxidative stress, it may act as a pro-oxidant by donating free copper ions, which induces reactive oxygen species (ROS) formation and low-density lipoprotein (LDL) oxidation [60].

In healthy human volunteers, significant decreases in catalase activity and ceruloplasmin levels by 37.38% and 32.97% were observed in sweetener consumers relative to non-consumers, respectively.

Similar results were also obtained for catalase activity which decreased by 12.38% in DS consumers, whereas ceruloplasmin level was increased by 13.17%. The elevated ceruloplasmin level could be related to its function as an acute phase protein as the ceruloplasmin level could be reflecting acute and chronic inflammation in an organism [60]. Elevated serum ceruloplasmin levels have been observed in type 2 diabetes [61].

As the daily amount of saccharin consumed increased from <5 to 5–10 and finally to >10 tablets/day, catalase activity decreased from 9.66 to 8.85 and to 6.97 kU/L, respectively. Similar results were obtained when the effect of the period of sweetener consumption was studied, A graded decrease was observed in catalase activity from 9.37 to 8.52 and to 8.44 kU/L as the consumption period increased from <5 to 5–10 to >10 years, respectively.

Finally, ceruloplasmin levels decreased from 30 to 26.37 to 22.52 mg/mL as the consumption increased from <5 to 5–10 to >10/day, respectively. Ceruloplasmin levels also decreased with prolonged sweetener consumption from 27.74 to 28.1 to 23.87 mg/mL, as the duration increased from <5 to 5–10 to >10 years, respectively.

Additional studies on the chronic use of artificial sweeteners where the diet of the participants is under control, including their consumption of sweetened foods and beverages are still needed.

## 5. Conclusions

We conclude that the prolonged consumption of saccharin and cyclamate mixtures induces oxidative stress in both healthy individuals and diabetic patients. The consumption of these artificial sweeteners reduces glycemic control, and leads to an elevated risk of atherosclerosis, in addition to inducing impairments in liver and kidney function. We find these effects to be exacerbated by increasing both dosage and duration of daily sweetener consumption.

## Figures and Tables

**Figure 1 medicina-59-00698-f001:**
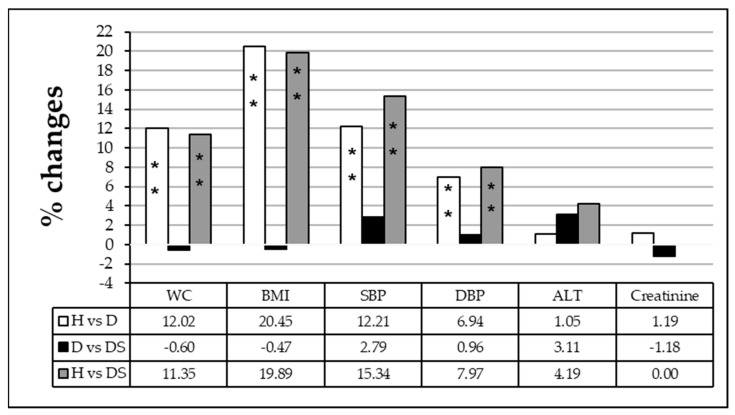
Percentage of change in waist circumference (WC), body mass index (BMI), systolic blood pressure (SBP), diastolic blood pressure (DBP), alanine transaminase (ALT) activity, and creatinine level among healthy individuals (H), diabetic patients (D) and diabetic patients sweetener consumers (DS). The comparison was performed using one-way ANOVA. Values presenting *p* > 0.05 were considered non-significant (NS). Significant values are represented by * *p* < 0.05 and ** *p* <0.001.

**Figure 2 medicina-59-00698-f002:**
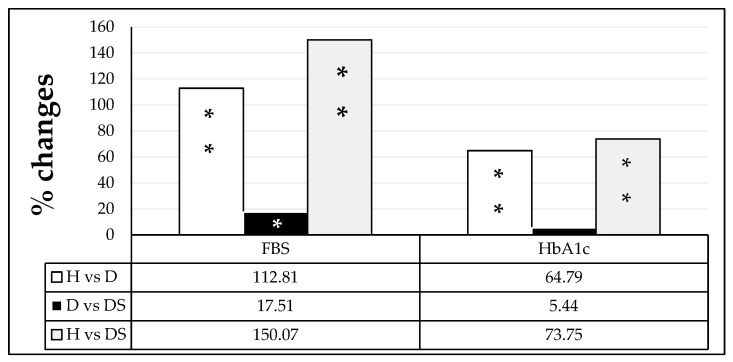
Percentage of change in fasting blood glucose (FSG) and hemoglobin A1C (HbA1C) among healthy individuals (H), diabetic patients (D), and diabetic patients sweetener consumers (DS). The comparison was performed using one-way ANOVA. Values presenting *p* > 0.05 were considered non-significant (NS). Significant values are represented by * *p* < 0.05 and ** *p* < 0.001.

**Figure 3 medicina-59-00698-f003:**
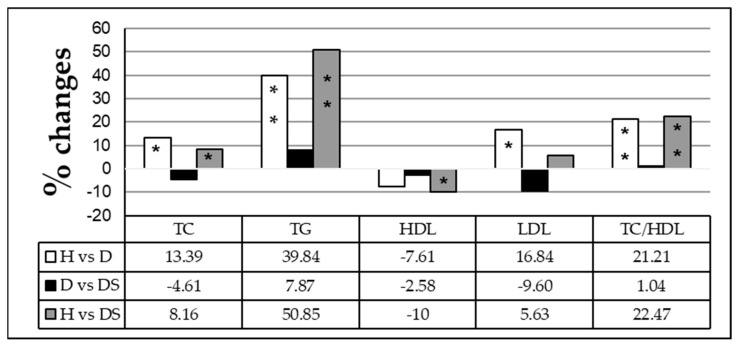
Percentage of change in lipid profile parameters among healthy individuals (H), diabetic patients (D), and diabetic patients sweetener consumers (DS). The comparison was performed using one-way ANOVA. Values presenting *p* > 0.05 were considered non-significant (NS). Significant values are represented by * *p* < 0.05 and ** *p* <0.001.

**Figure 4 medicina-59-00698-f004:**
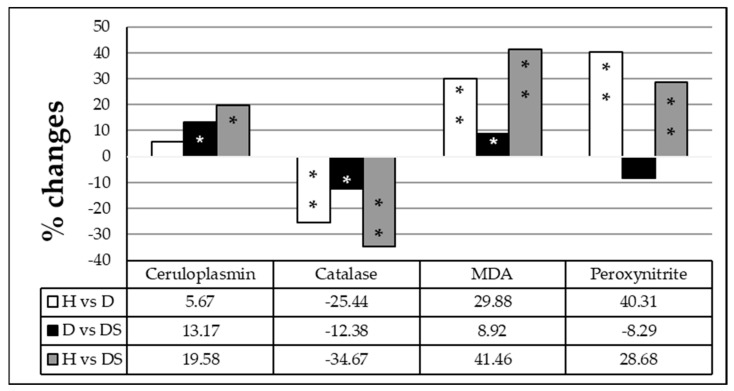
Percentage of change in the oxidative stress parameters ceruloplasmin, catalase, and malondialdehyde (MDA) among healthy individuals (H), diabetic patients (D), and diabetic patients sweetener consumers (DS). The comparison was performed using one-way ANOVA. Values presenting *p* > 0.05 were considered non-significant (NS). Significant values are represented by * *p* < 0.05 and ** *p* < 0.001.

**Figure 5 medicina-59-00698-f005:**
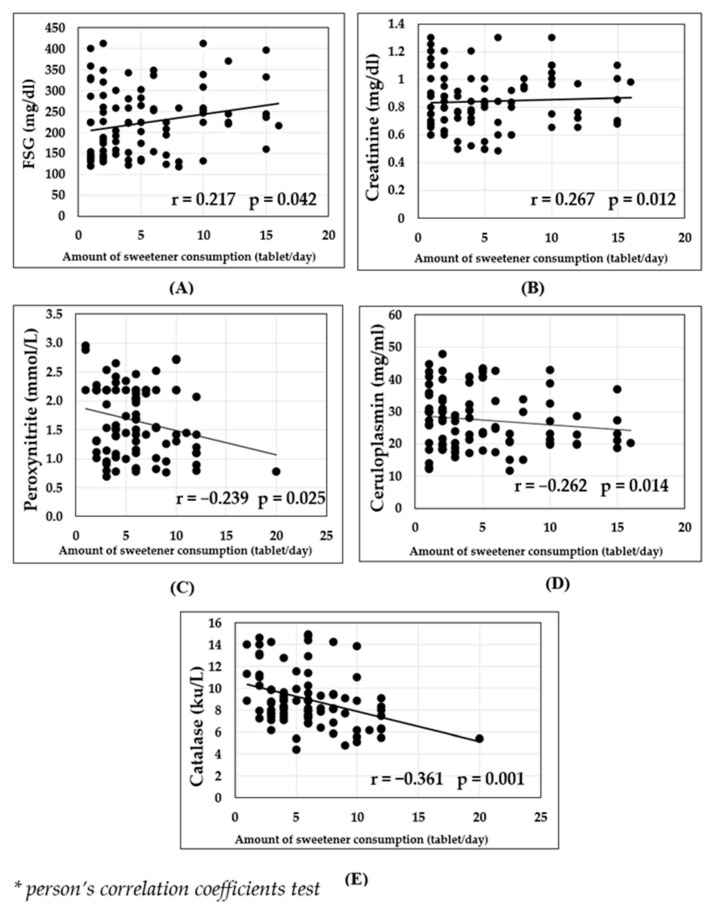
Pearson Correlation Coefficient (r) was calculated to analyze if there was a linear correlation between daily sweetener consumption and the concentration of the following parameters in diabetic patients: fasting blood glucose (**A**), creatinine (**B**), peroxynitrite (**C**), serum ceruloplasmin (**D**) and activity of serum catalase (**E**).

**Figure 6 medicina-59-00698-f006:**
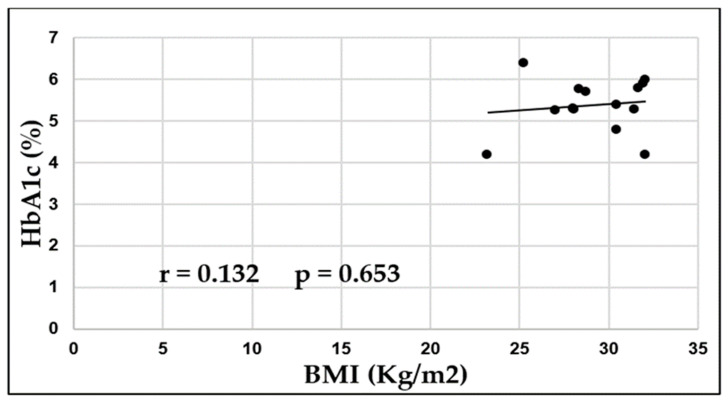
Pearson Correlation Coefficient (r) was calculated to analyze if there was a linear correlation between BMI (Kg/m^2^) and HbA1c (%) among healthy individuals sweetener consumers.

**Figure 7 medicina-59-00698-f007:**
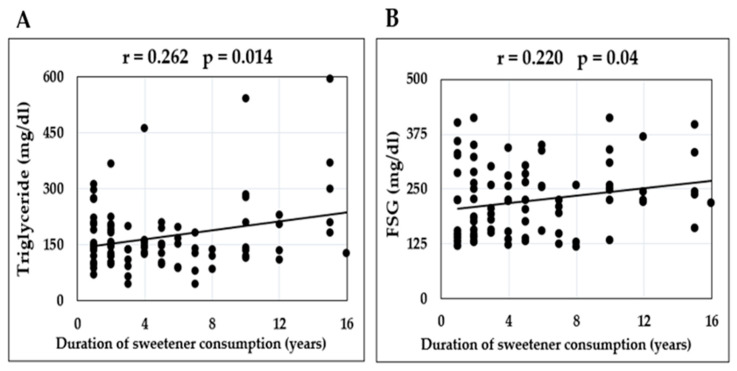
Pearson Correlation Coefficient was calculated to analyze if there was a linear correlation between the duration of daily sweetener consumption and concentration of the following parameters in diabetic patients: serum triglycerides (**A**) and fasting blood glucose (**B**).

**Table 1 medicina-59-00698-t001:** Effect of artificial sweetener consumption among diabetic patients.

Variables	T2DM without Sweeteners (D) (n = 93) (Mean ± SD)	T2DM with Sweeteners (DS) (n = 88) (Mean ± SD)	*p* * Value
WC (cm)	106.04 ± 7.91	105.40 ± 9.83	NS
BMI (Kg/m^2^)	31.80 ± 4.49	31.65 ± 4.78	NS
Systolic BP (mm Hg)	136.13 ± 15.18	139.93 ± 16.67	NS
Diastolic BP (mm Hg)	85.32 ± 6.79	86.14 ± 7.46	NS
FSG (mg/dL)	189.91 ± 60.55	223.16 ± 80.56	0.002
HbA1c (%)	7.91 ± 1.6	8.34 ± 1.71	NS
ALT (IU/L)	12.53 ± 4.78	12.92 ± 4.62	NS
Creatinine (mg/dL)	0.85 ± 0.19	0.84 ± 0.21	NS
Total Cholesterol (mg/dL)	194.91 ± 53.07	185.92 ± 44.51	NS
Triglyceride (mg/dL)	159.54 ± 63.10	172.09 ± 94.60	NS
HDL (mg/dL)	42.23 ± 11.18	41.14 ± 11.59	NS
LDL (mg/dL)	120.78 ± 45.51	109.19 ± 37.55	NS
TC/HDL	4.80 ± 1.43	4.85 ± 1.76	NS
Ceruloplasmin (mg/mL)	24.23 ± 9.92	27.42 ± 9.14	0.025
Catalase (kU/L)	10.26 ± 4.66	8.99 ± 2.58	0.026
MDA (nmol/mL)	2.13 ± 0.65	2.32 ± 0.54	0.033
Peroxynitrite (mmol/L)	1.81 ± 0.59	1.66 ± 0.6	NS

The evaluation of the effect of artificial sweetener consumption among type 2 diabetic patients was performed by comparing diabetic patients who have consumed from 3 to 15 tablets of sweetener per day (DS) during at least one year with diabetic patients who were not sweetener consumers (D). The comparison was performed using One way ANOVA. Values presenting *p* > 0.05 were considered non-significant (NS).

**Table 2 medicina-59-00698-t002:** Effect of artificial sweetener consumption among healthy individuals.

Variables	Healthy without Sweeteners (H) (n = 68) (Mean ± SD)	Healthy with Sweeteners (HS) (n = 14) (Mean ± SD)	*p* * Value
WC (cm)	94.66 ± 7.7	100 ± 8.74	NS
BMI (Kg/m^2^)	26.4 ± 2.53	29.18 ± 2.75	0.001
Systolic B (mm Hg)	121.32 ± 8.67	123.57 ± 10.82	NS
Diastolic BP (mm Hg)	79.78 ± 6.32	80.71 ± 5.84	NS
FSG (mg/dL)	89.24 ± 11.95	90.71 ± 7.88	NS
HbA1c (%)	4.80 ± 0.44	5.38 ± 0.64	<0.001
ALT (IU/L)	12.4 ± 4.3	13.94 ± 4.39	NS
Creatinine (mg/dL)	0.84 ± 0.17	0.89 ± 0.14	NS
Total Cholesterol (mg/dL)	171.90 ± 38.68	185.43 ± 21.58	NS
Triglyceride (mg/dL)	114.09 ± 41.97	134.43 ± 60.34	NS
HDL (mg/dL)	45.71 ± 11.62	43.14 ± 8.05	NS
LDL (mg/dL)	103.37 ± 33.09	115.40 ± 17.45	NS
TC/HDL	3.96 ± 1.23	4.40 ± 0.76	NS
Ceruloplasmin (mg/mL)	22.93 ± 8.01	15.86 ± 2.98	0.002
Catalase (kU/L)	13.76 ±5.63	8.31 ± 1.72	0.001
MDA (nmol/mL)	1.64 ± 0.63	2.56 ± 0.32	<0.001
Peroxynitrite (mmol/L)	1.29 ± 0.55	1.17 ± 0.431	NS

The evaluation of the effect of artificial sweetener consumption among healthy individuals was performed by comparing healthy individuals who have consumed from 3 to 15 tablets of sweetener per day (HS) during at least one year with healthy individuals who were not sweetener consumers (H). The comparison was performed using One way ANOVA. Values presenting *p* > 0.05 were considered non-significant (NS).

**Table 3 medicina-59-00698-t003:** Change on anthropometric and biochemical parameters associated with the consumption of artificial sweeteners in healthy individuals.

Variables	H vs. HS	*p* * Value
WC (Cm)	5.64	0.049
BMI (Kg/m^2^)	10.53	0.003
SBP (mm Hg)	1.85	NS
DBP (mm Hg)	1.17	NS
FSG (mg/dL)	1.65	NS
HbA1c (%)	12.08	0.005
ALT (U/L)	12.42	NS
Creatinine (mg/dL)	5.95	NS
TC (mg/dL)	7.87	NS
TG (mg/dL)	17.83	NS
HDL (mg/dL)	−5.62	NS
LDL (mg/dL)	11.64	NS
TC/HDL	11.11	NS
Ceruloplasmin (mg/mL)	−30.83	<0.001
Catalase (kU/L)	−39.61	<0.001
MDA (nmol/mL)	56.10	<0.001
Peroxynitrite (mmol/L)	−9.30	NS

Evaluation of the percentage of change on demographic and biochemical parameters associated with the consumption of artificial sweeteners in healthy individuals sweetener consumers (HS) and in healthy individuals (H) not sweetener consumers. All the individuals included in the group as sweetener consumers have consumed sweeteners for more than one year. The comparison was performed using One way ANOVA. Values presenting *p* > 0.05 were considered non-significant (NS).

**Table 4 medicina-59-00698-t004:** Impact of the amount of sweetener consumed for long term among type 2 diabetic patients.

Variables	Amount of Sweetener Consumption for ≥1 Year in T2DM Patients			*p* * Value
	<5 Tablets/Day	5–10 Tablets/Day	>10 Tablets/Day	
	(N = 35)	(N = 44)	(N = 9)	
WC ** (cm)	105.3 ± 9.67	105.84 ± 10.1	104.67 ± 10.08	NS
BMI ** (Kg/m^2^)	31.9 ± 4.72	31.59 ± 5.06	30.97 ± 3.92	NS
Systolic BP ** (mm Hg)	137.71 ± 16.73	142.39 ± 16.76	136.56 ± 16.02	NS
Diastolic BP **(mm Hg)	85 ± 9.16	87.16 ± 6.32	85.56 ± 5.27	NS
FSG ** (mg/dL)	196.97 ± 69.89	242.5 ± 86.27	230.44 ± 68.53	0.041
HbA1c ** (%)	7.98 ± 1.69	8.53 ± 1.66	8.85 ± 1.93	NS
ALT ** (IU/L)	13.6 ± 4.94	12.63 ± 4.21	11.66 ± 5.36	NS
Creatinine ** (mg/dL)	0.76 ± 0.19	0.9 ± 0.19	0.87 ± 0.25	0.008
Total Cholesterol ** (mg/dL)	186.4 ± 4.07	189 ± 50.17	169 ± 28.61	NS
Triglyceride ** (mg/dL)	155.77 ± 60.19	193.5 ± 116.88	130.89 ± 53.88	NS
HDL ** (mg/dL)	41.86 ± 12.11	40.61 ± 11.92	40.89 ± 8.36	NS
LDL ** (mg/dL)	108.66 ± 38.65	111.65 ± 38.52	99.22 ± 29.33	NS
TC/HDL **	4.89 ± 1.96	4.92 ± 1.69	4.31 ± 1.25	NS
MDA ** (nmol/mL)	2.46 ± 0.57	2.21 ± 0.51	2.29 ± 0.51	NS
Ceruloplasmin ** (mg/mL)	30 ± 5.54	26.37 ± 9.28	22.52 ± 8.62	0.049
Peroxynitrite ** (mmol/L)	1.74 ± 0.65	1.69 ± 0.56	1.2 ± 0.41	0.052
Catalase ** (kU/L)	9.66 ± 2.37	8.85 ± 2.72	6.97 ± 1.30	0.015

Evaluation of the impact of the amount of sweetener consumed for a long term among type 2 diabetic patients on anthropometric and biochemical parameters. The analysis was performed by comparing diabetic patients who have consumed less than 5 tablets a day, 5–10 tablets a day, and more than 10 tablets a day. The comparison was performed using one-way ANOVA. Values presenting *p* > 0.05 were considered non-significant (NS).

**Table 5 medicina-59-00698-t005:** Impact of artificial sweeteners long-term consumption among type 2 diabetic patients.

Variables	Duration of Sweetener Consumption ≥1 Tablet/Day in Diabetic Patients			*p* * Value
	<5 Years	5–10 Years	>10 Years	
	(N = 49)	(N = 29)	(N = 10)	
WC (cm)	106.35 ± 10.89	103.34 ± 7.9	106.7 ± 9.35	NS
BMI (Kg/m^2^)	32.42 ± 4.93	30.71 ± 4.39	30.61 ± 4.9	NS
Systolic BP (mm Hg)	138.55 ± 16.49	140.72 ± 16.27	141.5 ± 19.73	NS
Diastolic BP (mm Hg)	86.94 ± 7.89	85.34 ± 6.4	84.50 ± 8.32	NS
FSG (mg/dL)	211.82 ± 81.87	228.38 ± 77.54	263.6 ± 75.37	NS
HbA1c (%)	8.07 ± 1.74	8.65 ± 1.77	8.8 ± 1.19	NS
ALT (IU/L)	13.55 ± 4.43	12.22 ± 5.24	11.86 ± 3.32	NS
Creatinine (mg/dL)	0.84 ± 0.21	0.84 ± 0.22	0.84 ± 0.16	NS
Total Cholesterol (mg/dL)	188.2 ± 42.47	174.62 ± 47.54	207.5 ± 39.29	NS
Triglyceride (mg/dL)	164.9 ± 77.91	158.86 ± 91.42	245.7 ± 145.84	0.03
HDL (mg/dL)	40.33 ± 9.39	44 ± 14.96	36.8 ± 8.95	NS
LDL (mg/dL)	111.82 ± 35.76	97.67 ± 37.83	129.72 ± 37.72	0.049
TC/HDL	4.88 ± 1.43	4.38 ± 1.95	6.05 ± 2.22	0.033
MDA (nmole/mL)	2.37 ± 0.60	2.26 ± 0.40	2.21 ± 0.59	NS
Ceruloplasmin (mg/mL)	27.74 ± 9.11	28.10 ± 10.10	23.87 ± 5.6	NS
Peroxynitrite (mmole/L)	1.67 ± 0.61	1.58 ± 0.59	1.86 ± 0.60	NS
Catalase (kU/L)	9.37 ± 2.51	8.52 ± 2.79	8.44 ± 2.12	NS

The evaluation of the impact of artificial sweetener long-term consumption among type 2 diabetic patients was performed by comparing patients who consumed sweeteners for 1–4 years, 5–10 years, and more than 10 years. The comparison was performed using one-way ANOVA. Values presenting *p* > 0.05 were considered non-significant (NS).

## Data Availability

The datasets generated and/or analyzed during the current study are not publicly available as they were generated from samples of human material obtained through a specific consent form, but are available from the corresponding author upon reasonable request.

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
