# Peer review of "Impact of Long-Term Cyclamate and Saccharin Consumption on Biochemical Parameters in Healthy Individuals and Type 2 Diabetes Mellitus Patients"

_medicina, 2023, doi:10.3390/medicina59040698_

Round 1

Reviewer 1 Report

This scientific research work deals with a very interesting topic of importance for the entire social population, both healthy and people suffering from diabetes mellitus type 2, which is still insufficiently reached and researched on humans and whose results could be important as a initial point for future expressions and a possible turning point in the current attitudes about the usage of artificial sweeteners. Research is very important  for society as well as for the academic community.

I would have a few suggestions before publication, so that the work would gain even more importance. 

The first suggestion concerns about the design of the study and I would ask the author to modify it according to the guidelines of the journal- Abstract, Keywords, Introdcution, Material and Methods, Results, Discussion, Conclusion, References, Tables and Graphs in the end. 

Secondly, the statistics of the work is based on independent t-test and one-way ANOVA of which application the prerequisite is the normal distribution (CV), which is not specified anywhere in this paper for any group of respondents. Calculate and note it. 

Then, the results indicate that systolic and diastolic pressure is elevated in both groups of diabetics compared to the healthy group. This data is not relevant for the reason that people with type 2 diabetes, who are also obese, as stated above, most likely developed the clinical picture of metabolic syndrome, which therefore indicates that they have elevated blood pressure values. Also, FSG, HbA1c, TC, TG, LDL, TC/HDL results presented a significant increase in group D compared to group H. That is as expected. These comparisons are also inadequate. If we want to compare the lipid status, glycemic status, as well as the oxidative status of healthy people and diabetics, who used artificial sweeteners or not, in that way it is necessary to determine their base glycemic, lipid an oxidative status. In that case, if these parameters are similar in the beginning in both groups, without statistical significance, then the comparison and obtained results would be adequate.

Author Response

Dear reviewer 1,

first of all, we would like to thank you for your comments and suggestions.  Your comments were highly appreciated.

Below are the responses to your comments and suggestions.

Thank you.

Answers to the reviewer 1 comments and suggestions:

Reviewer 1:

1- The first suggestion concerns about the design of the study and I would ask the author to modify it according to the guidelines of the journal- Abstract, Keywords, Introduction, Material and Methods, Results, Discussion, Conclusion, References, Tables and Graphs in the end. 

Answer: the paper was now corrected following the guidelines of the journal. Now the template offered by the journal was applied.

2- Secondly, the statistics of the work is based on independent t-test and one-way ANOVA of which application the prerequisite is the normal distribution (CV), which is not specified anywhere in this paper for any group of respondents. Calculate and note it. 

Answer: all the study groups were analyzed regarding their normal distribution    by Skewness and Kurtosis Test before proceeding the comparisons  between the studied groups. This information was included in “Material and Methods”, subitem “2.5 Statistical analysis”.

3- Then, the results indicate that systolic and diastolic pressure is elevated in both groups of diabetics compared to the healthy group. This data is not relevant for the reason that people with type 2 diabetes, who are also obese, as stated above, most likely developed the clinical picture of metabolic syndrome, which therefore indicates that they have elevated blood pressure values. Also, FSG, HbA1c, TC, TG, LDL, TC/HDL results presented a significant increase in group D compared to group H. That is as expected.

Answer: yes, the result obtained regarding blood pressure, FSG, HbA1c, TC, TG, LDL, TC/HDL is indicating that the obese diabetic participants have developed metabolic syndrome. This result suggests that perhaps the use of sweeteners does not help to prevent, nor does it improve the metabolic syndrome. In this matter, our data are reinforcing previous findings.

4- These comparisons are also inadequate. If we want to compare the lipid status, glycemic status, as well as the oxidative status of healthy people and diabetics, who used artificial sweeteners or not, in that way it is necessary to determine their base glycemic, lipid an oxidative status. In that case, if these parameters are similar in the beginning in both groups, without statistical significance, then the comparison and obtained results would be adequate.

Answer: unfortunately, we did not have access to the baseline levels of the parameters evaluated for each group, as the sweetener users were already using sweeteners when the study started. One of the main goals of the study was to evaluate the chronic effect of artificial sweeteners on the health of the individuals. Because of that, all the individual’s sweetener users were already using sweetener. This was not an interventional study. However, we have healthy and diabetic individuals who were not using sweetener and that was the best possibility we had to make the comparisons between the groups.

Reviewer 2 Report

This manuscript investigated the effects of chronic consumption of saccharin and cyclamate on biochemical parameters in healthy individuals and patients with type 2 diabetes mellitus.

This is an interesting study.

My comments are the following:

The introduction does not include the purpose of the study.

It is not clear how much saccharin and cyclamate patients consumed in a day. / in mg/bw/ Have patients/ healthy participants met or exceeded the recommended ADI?

There is a big difference between taking 3 sweetener tablets a day or 15 tablets a day. Not taking this can lead to bias.

Did the authors investigate whether participants had consumed the same sweetener for 5 or 10 years, or whether they regularly consumed other sweetener-containing foods/drinks/beverages? The lack of this also biases the results.

The authors did not investigate whether there was a relationship between BMI and HbA1c levels in healthy participants who consumed sweeteners. It is not clear why the authors think that the significant difference in HbA1c is caused by the regular consumption of sweeteners in healthy people.

The authors explained the relationship between sweeteners and HbA1c, but did not examine dietary adherence among people with diabetes.

You mentioned that” Chronic consumption of saccharin may cause kidney injury”, but it is not only the length of time that is important, but also the amount consumed, which is not clear from your study.

You mentioned that „Consumption of sweeteners in healthy individuals led to a significant raise in triglyceride (16.74%), LDL (13.39%), and TC/HDL (13.11%)”- but the authors did not study the diets, lifestyles or other risk factors of healthy people, so it is risky to claim this.

In conclusion, you mentioned „The consumption of these artificial sweeteners increases the risk of obesity”- How did the authors determine this? Both healthy groups were overweight according to BMI values. This may have been the reason why 14 people started using sweeteners. It is therefore unlikely that sweeteners caused the overweight.  

The novel of your study is the effects of sweeteners on oxidative stress  creatinine and alanine transaminase activity.

The references used are not up to date and there are many new articles on this topic.

Author Response

Dear reviewers 2,

first of all, we would like to thank you for your comments and suggestions.  Your comments were highly appreciated.

Below are the responses to your comments and suggestions.

Thank you.

Answers to the reviewers 2’ comments and suggestions:

Reviewer 2:

1- The introduction does not include the purpose of the study.

Answer: the purpose of the study was now included in the last paragraph of the introduction.

2- It is not clear how much saccharin and cyclamate patients consumed in a day. / in mg/bw/ Have patients/ healthy participants met or exceeded the recommended ADI? There is a big difference between taking 3 sweetener tablets a day or 15 tablets a day. Not taking this can lead to bias.

Answer: now the description of the concentration of cyclamate and saccharin in the tablets was included in “item 2.2 Study group” in Material and Methods. All sweetener-consuming participants used doses within the ADI. This information was also now included in Material and Methods.

3- Did the authors investigate whether participants had consumed the same sweetener for 5 or 10 years, or whether they regularly consumed other sweetener-containing foods/drinks/beverages? The lack of this also biases the results.

Answer: the participants used the same type of sweetener throughout the time considered in the study. Unfortunately, we did not have access to information regarding the consume of sweeter-containing foods, drinks or beverages. The observation that there still a need for additional studies in humans where the chronic consume of sweetened-foods, drinks and beverages would be under control as well as the tablets of artificial sweeteners, was included in the last paragraph of the discussion.

4- The authors did not investigate whether there was a relationship between BMI and HbA1c levels in healthy participants who consumed sweeteners. It is not clear why the authors think that the significant difference in HbA1c is caused by the regular consumption of sweeteners in healthy people. The authors explained the relationship between sweeteners and HbA1c, but did not examine dietary adherence among people with diabetes.

Answer: now the relationship between BMI and HbA1 in healthy individuals was investigated using Pearson Correlation Coefficient (r) calculation. A linear but non-significant correlation was found between BMI and HbA1. The analysis was added to the manuscript (Figure 6).

5- You mentioned that” Chronic consumption of saccharin may cause kidney injury”, but it is not only the length of time that is important, but also the amount consumed, which is not clear from your study.

Answer: we totally agree. Now the information regarding the concentration of cyclamate and saccharin used was added, as mentioned before. The effect of the amount and the time of sweetener consumption is showed in Table 4 and 5 and the results were discussed accordingly. We found that the effect of chronic use of the studied artificial sweeteners is both, time and concentration dependent. It was also mentioned in the conclusion.

6- You mentioned that „Consumption of sweeteners in healthy individuals led to a significant raise in triglyceride (16.74%), LDL (13.39%), and TC/HDL (13.11%)”- but the authors did not study the diets, lifestyles or other risk factors of healthy people, so it is risky to claim this.

Answer: in the abstract that statement was now replaced by “The results suggest that saccharin and cyclamate increased HbA1C (+11.16%), MDA (+52.38%), TG (+16.74%), LDL (+13.39%) and TC/HDL (+13.11%) in healthy volunteers.” Also the discussion was modified in the item “4.4 Effect of artificial sweeteners consumption on lipid profile”.

7- In conclusion, you mentioned „The consumption of these artificial sweeteners increases the risk of obesity”- How did the authors determine this? Both healthy groups were overweight according to BMI values. This may have been the reason why 14 people started using sweeteners. It is therefore unlikely that sweeteners caused the overweight.  

Answer: Now all participants with obesity have been excluded from the healthy group (H). Comparisons using the healthy group were all recalculated and the results were updated accordingly. Because of that, some results became highly significant (p<0.001) as highlighted in the text. The statement saying that the use of cyclamate and saccharin increase the risk of obesity was now removed from the conclusion.

8- The references used are not up to date and there are many new articles on this topic.

Answer: the references were updated accordingly. The updated references are highlighted in the text.

Round 2

Reviewer 2 Report

This is the second time I have reviewed this document. I am pleased to see that the manuscript has improved significantly since my first review and that many of my comments have been incorporated into the current draft. I therefore recommend the manuscript for acceptance.